# The intake of ultra-processed foods, all-cause, cancer and cardiovascular mortality in the Korean Genome and Epidemiology Study-Health Examinees (KoGES-HEXA) cohort

**Anthony Kityo**[1], **Sang-Ah Lee**[1,2]*

**1** Department of Preventive Medicine, School of Medicine, Kangwon National University, Chuncheon, Gangwon, Republic of Korea, **2** Interdisciplinary Graduate Program in Medical Bigdata Convergence, Kangwon National University, Chuncheon, Gangwon, Republic of Korea

* sangahlee@kangwon.ac.kr

## Abstract

The relationship between ultra-processed food (UPF) intake and mortality is unknown in Asian countries, yet the intake of UPF is on the rise in these countries. This study examined the association of UPF intake with all-cause, cancer and cardiovascular disease (CVD) mortality. Participants were 113,576 adults who responded to a 106-item food frequency questionnaire during recruitment of the 2004–2013 Health Examinees (HEXA) study, a prospective cohort study in Korea. UPF were defined using the NOVA classification and evaluated as quartiles of the proportion of UPF in the diet (% total food weight). Multivariable Cox regression and restricted cubic spline models were used to examine the association of UPF intake with all-cause and cause specific mortality. A total of 3456 deaths were recorded during a median follow-up of 10.6 (interquartile range, 9.5–11.9) years. There was no evidence of an association of UPF intake with all-cause, cancer or CVD mortality comparing the highest with the lowest quartiles of UPF intake (all-cause mortality, men: hazard ratio [HR] 1.08, 95% confidence interval [CI] 0.95–1.22; women: HR 0.95, 95% CI 0.81–1.11; cancer mortality, men: HR 1.02, 95% confidence interval [CI] 0.84–1.22; women: HR 1.02, 95% CI 0.83–1.26; CVD mortality, men: HR 0.88, 95% CI 0.64–1.22; women: HR 0.80, 95% CI 0.53–1.19). However, the risk of all-cause mortality increased in both men and women with high consumption of ultra-processed red meat and fish (men, HR 1.26, 95% CI 1.11–1.43); women, HR 1.22 95% CI 1.05–1.43); and in men with high consumption of ultra-processed milk (HR 1.13, 95% CI 1.01–1.26); and soymilk drink (HR 1.12, 95% CI 1.00–1.25). We found no evidence of an association between total UPF intake and all-cause, cancer or CVD mortality, but ultra-processed red meat and fish in both sexes, and milk and soymilk drinks in men were positively associated with all-cause mortality.

**Data Availability Statement:** Data from the Health Examinees (HEXA) study is part of the Korean Genome and Epidemiology Study (KoGES), conducted by Korea Disease Control and

Prevention Agency (KDCA). The Health Examinees Study dataset used in our study was merged with the Central Cancer Registry (KCCR) data provided by National Cancer Center of Korea in a collaborative agreement. The dataset analyzed in this study is maintained and managed by the Division of Population Health Research at the National Institute of Health, Korean Disease Control and Prevention Agency. It contains personal data that may potentially be sensitive to the patients, even though researchers are provided with an anonymized dataset that excludes resident registration numbers. Accordingly, the minimal data set used in the current study could not be publicly shared by the authors due to legal restriction on sharing sensitive patient information. Researchers are required to submit ethics approval, and a detailed research plan to the KDCA. Upon approval, the researchers are required to physically visit the KCDA and conduct the analysis from the KoGES data analysis room at the KCDA in Osong, Chungcheong Province, Republic of Korea. However, if the analysis does not involve linkage to the cancer registry, virtual access to the anonymized data set can be granted. Other researchers may request access to the anonymized data by contacting the following individuals at the Division of Population Health Research, National Institute of Health, Korea Disease Control and Prevention Agency: Senior Staff Scientist Dr. Jung Hyun Lee (jaylee1485@korea.kr); Director Dr. Kyoungho Lee (khlee3789@korea.kr).

**Funding:** The authors received no specific funding for this work.

**Competing interests:** The authors have declared that no competing interests exist.

**Abbreviations:** AMI, Activity Metabolic Index; BMI, body mass index; CI, confidence interval; CHD, coronary heart disease; CKD, chronic kidney disease; CVD, cardiovascular disease; eGFR, estimated glomerular filtration rate; HEXA, health examinees cohort; HR, hazard ratio; KoGES, Korean Genome and Epidemiology Study; LTPA, leisure time physical activity; RDA, Rural Development Administration; SQFFQ, semi-quantitative food frequency questionnaire; TFBCs, transnational food and beverage corporations; UPF, ultra-processed foods.

# Introduction

Global dietary patterns are changing and tending towards the consumption of ultra-processed foods (UPF). Owing to their dietary and chemical composition including food additives, and the method used to process these foods, the dietary quality of UPF and their contribution to dietary intake and health outcomes have become a center of attention in recent years. UPF are high in energy, sugars, and trans-fat, and are deficient in fiber, various micronutrients, and other bioactive compounds [1] The consumption of these foods has been linked to cardiovascular disease (CVD), coronary heart disease (CHD), cerebral vascular disease [2, 3], and cancer [4].

A recent systematic review found a positive link between UPF consumption and various health outcomes including mortality, hypertension, mental disorders, metabolic syndrome, respiratory disorders, and obesity [5]. A cross-sectional study in Korea found a positive association between UPF and hypertension [6]. Non-communicable diseases are the leading causes of premature mortality and long-lasting disability [7]. Previous studies have investigated the longitudinal association of UPF intake with total and cause-specific mortality. A positive association between UPF intake and all-cause or cause-specific mortality has been reported in the North America [8, 9], Spain [10–12], France [13] and Italy [14, 15]. There is need to replicate these finding in other regions such as Asia which have different consumption and processing patterns of UPF and report significant mortality rates associated with CVD and cancer.

Asian countries are currently under the influence of transnational food and beverage corporations (TFBCs) and the consumption of UPF in the Asian region is increasing. TFBCs alter the availability, price, dietary quality, desirability, and consumption of UPF [16]. Although the consumption of UPF such as sugar-sweetened beverages and processed red meats is low, Korea has experienced a gradual increase in the energy contribution of UPFs to the total diet [17]. The proportion of energy from UPFs in Korea is reported at 25.1% [18], which is comparable to that in developing countries where UPF intake has been linked to an increased risk of mortality. In addition, UPF intake is associated with hypertension in Korea, the most important risk factor for CVD [6]. Thus, research on the link between UPF consumption and mortality in the Korean population is warranted.

Moreover, UPF intake in Korea is high in men, those who live in urban areas, the highly educated, the young and those with high income [17]. Therefore, it is necessary to evaluate the moderating roles of demographic, lifestyle, and socioeconomic factors in the association between UPF consumption and mortality risk. This study aimed to investigate the association of UPF intake with all-cause, cancer and CVD mortality. Our study aimed to highlight the role of food processing in the primary prevention of premature mortality and to replicate previous findings in an Asian population that is experiencing a gradual increase in the consumption of UPF and has dietary patterns that are different from those in Western populations.

# Materials and methods

## Study population

The Health Examinees (HEXA) cohort is a prospective population-based study within the Korean Genome and Epidemiology Study (KoGES), a nationwide study that was established to investigate the etiological factors of complex diseases [19]. The HEXA recruited participants between 2004–2013 at 38 health examination centers and training hospitals located in the eight regions of Korea. The HEXA study continues to follow up all participants in accordance with a standardized study protocol. The followed-up participants are periodically invited to complete the surveys by mail and telephone calls. The study details have been published elsewhere [20].

For this analysis, anonymized data from 173,202 participants $\geq$ 40 years were combined with the death certificate database of the National Statistical Office. Of these, 130,224 consented to be linked to the National death registry. We excluded participants from invalid recruitment sites: 1) sites that participated in the pilot study between 2004 and 2006; 2) sites that did not meet the HEXA standards for biospecimen quality control; 3) sites that participated in the study for less than two years [21] (n = 11,688); and 4) those aged less than 40 years or more than 69 years because they were not included in the original recruitment criteria of HEXA (n = 1848). Thus, the HEXA-Gem sample was obtained, which comprised 116,688 participants (Fig 1). We further excluded participants who had missing data on dietary intake (n = 1379) and those who reported implausible energy intake ($<$800 or $\geq$4000 kcal/day in men and $<$500 or $\geq$3500 kcal/day in women, n = 1733), yielding a final sample of 113,576 participants, of whom 74,729 were women and 38,847 were men.

The HEXA study was approved by the Ethics Committee of the Korean Health and Genomic Study of the Korean National Institute of Health and the Institutional Review Boards of all the participating hospitals (IRB no. E-1503-103–657). The present analysis was approved by the Institutional Review Board of Kangwon National University Hospital (IRB no. KWNUIRB-2022-07-007). All participants provided written informed consent prior to participating in the HEXA study. All the data were anonymized, and no individual participants could be identified.

## Measures

**Assessment of dietary intake.** Dietary intake was assessed once at baseline using a 106-food item semiquantitative food frequency questionnaire (SQFFQ). The SQFFQ was tested for reproducibility and validity using 12-day dietary records obtained from 124 participants [22]. Food consumption frequencies were classified into nine levels (from "never" to "three times or more a day"), and portion sizes were classified into three levels (one-half, one, and one and a half servings). Energy and macronutrient intake attributable to each food item were estimated using a food composition table developed by the Korean Rural Development Administration (RDA) [23].

We classified food items based on the degree and extent of processing using the NOVA classification [24]. We employed with slight modification a four-stage approach developed by Khandpul and colleagues to assign each food item to a NOVA category [25]. In brief, AK (nutritionist) independently assigned each of the 106 items to a NOVA group and then consulted a registered dietitian (SAL) who also participated in the design of the HEXA SQFFQ, to validate AK's classification. Regarding foods for which consensus could not be reached, AK visited food stores and checked websites to verify the food labelling information and manufacturing processes, respectively. We also referred to previous publications and checked their UPF categorizations [1, 17]. For mixed dishes or aggregated food groups which contained food items with different degrees of processing, we disaggregated them and applied weights using the Korean food recipe information. The applied weights represent the percentage of weight contributed by a food item to the food group or mixed dish. Finally, the UPF items in this study comprised instant noodles, breakfast cereals, breads, bread spreads (Jam, butter, margarine), cakes, cookies, crackers, snacks, candies, chocolate, pizza, and hamburgers, processed red meat (ham, sausage), processed fish (crab sticks), flavored milk, yoghurts, ice cream, soymilk drink, soft drinks and fruit sodas, and sweet rice punch (*Sikhye*) (S1 Table).

**Demographic, lifestyle, and physical characteristics.** Baseline characteristics and disease history were collected using a standardized interviewer-administered questionnaire. For descriptive purposes, participant characteristics were classified as follows: age (40–49, 50–59,

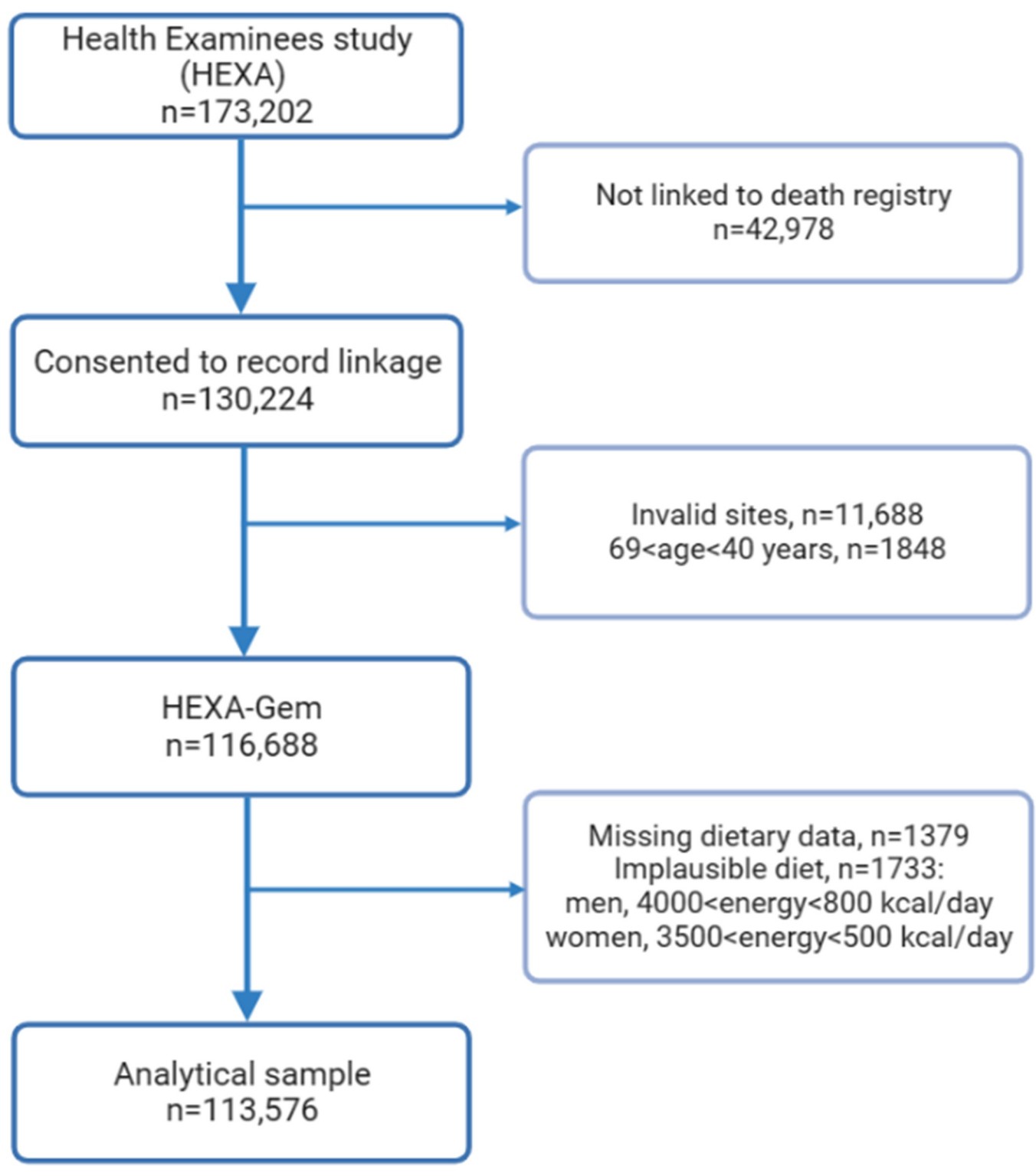

**Fig 1. Flowchart showing the selection of study participants.**

and 60–69 years); marital status (married/cohabiting, single/separated/divorced/widowed/others); educational level (≤elementary school, middle school, high school, and ≥university), and monthly family income (<1,000, 1,000–3000 and ≥3,000 USD). Smoking status was classified using the definition from a previous study [26]. Current smokers were defined as participants who had smoked >400 cigarettes in their lifetime and were still smoking at the time of the interview. Participants were categorized as current, past,s and never drinkers. Current alcohol drinkers were those who reported that they had ever drunk alcohol and were still drinking at the time of the interview. The daily intake of alcohol (g/day) was also computed for each participant using detailed alcohol assessment information from the questionnaire.

Regarding physical exercise, participants were asked to report 1) whether they engaged in regular physical exercise that causes body sweating; 2) the number of times they engaged in these exercises in a week (1–2 times/week to every day); and 3) the duration of exercise. Regular exercise was defined as engaging in activities that caused body sweating at least five times a week lasting at least 30 minutes per session. Leisure time physical Activity (LTPA) was assessed using the Taylor's Minnesota LTPA questionnaire. The Activity Metabolic Index (AMI) for light (<4.0 metabolic equivalent of task (METs)), moderate (4.0–5.5 METs) and high-intensity activities (>5.5 METs) was calculated [27]. Sedentary lifestyle was defined as moderate-intensity activity AMI of <675 kcal/week or high-intensity activity AMI<420 kcal/week [28]. Weight and height were objectively measured by trained medical staff at baseline. Body mass index (BMI) was calculated as weight in kilograms divided by the square of height in meters (kg/m$^2$). BMI was categorized into four classes based on the WHO classification for Asian adults:<18.5, 18.5–22.9, 23–24.9 and ≥25 kg/m$^2$ [29]. Anemia was defined as hemoglobin <13.0 g/dl in men and <12.0 g/dl in women. Elevated alanine transaminase (ALT) was defined as ALT>40 U/L). Hypercholesterolemia was defined as total cholesterol levels>240 mg/dl [30].

*Prevalence of chronic diseases*. Information about diseases and medication use was obtained using a standardized questionnaire administered by trained staff. The prevalence of cancer, myocardial information, ischemic stroke, chronic pulmonary disease, chronic gastritis, and diabetes at baseline was identified from self-reported history of disease and current use of medication. Chronic kidney disease was diagnosed using estimated glomerular filtration rate (eGFR<60 mL/min/1.73m$^2$) that was estimated using the Chronic Kidney Disease Epidemiology Collaboration equation (CKD-EPI) [31]. Cancer was identified using the International Classification of Diseases codes, 10th revision (ICD-10) in the National cancer registry. Diabetes was defined as fasting blood glucose ≥126 mg/dl, drug treatment for elevated fasting blood glucose levels. Hypertension was defined as systolic blood pressure ≥130 mmHg, diastolic blood pressure ≥85 mmHg, or drug treatment for elevated blood pressure. Abdominal obesity was defined as waist circumference ≥90 cm for men and ≥80 cm for women [32]. The disease score was computed for each participant based on the number of chronic diseases. Participants with prevalent disease were assigned a score of 1 and zero (0) otherwise, with the exception of cancer where prevalent cases were assigned a score of six (6) for metastatic tumors and two (2) for non-metastatic tumors.

**Mortality ascertainment.** The dates and causes of death from 2004 to 31 December 2020, were ascertained through linkage to the death certificate database of the Korean National Statistical Office. The deaths of participants on Medicaid were ascertained through linkage to the National Health Insurance Service. Participants' unique identifiers were used to add mortality data from Statistics Korea. For cause-specific mortality, the International Classification of Diseases, 10th Revision was used. ICD-10 code C00-C97 and I00-I99 were used to classify cancer and CVD-specific deaths, respectively.

## Study design and statistical analyses

All data were analyzed using SAS software (version 9.4; SAS Institute Inc., Cary, NC, USA), and statistical significance was defined as P<0.05. We computed the percentage of UPF in the total diet (% food weight from UPF = weight of UPF (g/d) *100/ total food (g/d)) and categorized participants into sex-specific quartiles of the % of food weight from UPF. Regarding the analyses of UPF items/subgroups for which low intake was reported, participants were categorized into tertiles (of soymilk drink, breakfast cereals and snacks, candies, chocolate, and ice cream) or at the median (of pizza and hamburgers, bread spreads, and soft drinks/fruit drinks). The distribution of participant characteristics according to quartiles of UPF intake was described using percentages for categorical variables and least square means for continuous variables. For 8.7% of participants who had missing data on income, and 31.8% who had missing data on sedentary lifestyle (31.8%), a category of missing data was created and labelled 'unknown.' Missing data on education (1.0%), marital status (0.3%), smoking (0.3%), drinking (0.3%), physical exercise (<0.1%), menopausal status (0.52%), use of oral contraceptives (0.52), anemia (2.8%), and elevated ALT (2.9%) were replaced by the mode of each variable. Missing data on BMI were replaced by the median value of BMI (23.7).

The follow-up time for each participant was calculated from the date at which the interview was conducted until the date of death. Participants who did not experience the event were censored on December 31, 2020. We selected confounders based on predictors of mortality that have been published in previous studies [30, 33], and by evaluating multivariable Cox regression models. To examine the association of total UPF intake or UPF items/subgroups with all-cause or cause-specific mortality, sex-stratified hazard ratios (HRs) and 95% confidence intervals (CIs) were estimated using Cox proportional hazard models with follow-up time as a time-scale variable. The HRs were computed across quartiles (Q) of UPF intake using Q1 as the reference category. We constructed the models as follows: Model 1 was adjusted for age (continuous), and total energy intake (continuous); model 2 was adjusted for model 1, education level, monthly income, area of residence, marital status, smoking, drinking, and regular physical exercise. Model 3 was further adjusted for BMI (continuous), and Model 4 was representative of model 3 adjusted for comorbidity score, anemia, elevated ALT, menopausal status, and use of oral contraceptives. For CVD-related mortality, model 4 was further adjusted for hypercholesterolemia. The linearity in HRs was tested by modelling the median of each UPF quartile as a continuous variable and reading its P-value (P for trend). The Benjamini-Hochberg correction was used to calculate the False Discovery rate (FDR)-adjusted P-values for trend [34]. We implemented a SAS macro developed by Loic et al. [35] to examine the dose-response association between UPF intake and mortality risk. Restricted cubic splines were fitted with 3 knots at the 5th, 50th and 95th percentiles of the UPF distribution, and using the median of UPF intake (5.6%) as the reference value.

We conducted sensitivity analyses to test the robustness of our findings: 1) to assess the influence of overall dietary intake, we further adjusted model 4 for unprocessed/minimally processed food intake (% of food weight). 2) We adjusted our estimates for LTPA-derived sedentary lifestyle (unknown, sedentary, or active) to minimize residual confounding from physical activity. 3) Similarly, we adjusted model 4 for daily alcohol consumption in grams per day (continuous) to test for residual confounding from alcohol intake. 4) To examine the influence of the missing data treatment strategy on study estimates, we restricted the analysis to participants with complete covariate data. 5) To minimize reverse causation and latent period bias, we excluded participants who died before 5 years of follow-up. Finally, we excluded individuals who died of accidents or injuries. We explored whether demographic, socioeconomic, lifestyle and clinical variables modified the association of UPF intake with mortality. This approach

was also meant to filter out residual confounding from these variables. Thus, we stratified the analyses by age (40–49, 50–59 and 60–69), education (≤elementary school, middle school, high school, and ≥university), marital status (married, single), income (<1000, 1000–3000, and ≥3000 USD), smoking (current, past, and never smoker), drinking (current, past, and never drinker), regular physical exercise (yes and no), BMI category (<18.5, 18.5–22.9, 23–24.9, and ≥25 kg/m$^2$), disease score (0, ≥1), postmenopausal status (yes, no) and use of oral contraceptives (yes and no). We computed the P- value for interaction by including the cross-product terms of the stratification variable and UPF in the model and reading the P-value of the cross-product term. The P values for interaction were also adjusted for FDR using the Benjamini-Hochberg correction.

## Results

This study analyzed 113,576 participants of whom 74,729 were women. The mean age (mean ± standard error) of participants was 52.0±0.1 years (53.7±0.1 in men and 52.4±0.1 in women), and the median UPF intake (% food weight) was 5.6% (5.9% in men and 6.2% in women). After 1,205,320 person years of follow-up (median (interquartile range) per participant of 10.6 (9.5–11.9 years), 3454 deaths were ascertained. The highest consumers of UPF were more likely to be younger, highly educated, have high income, and reside in Seoul and Metropolitan areas. In addition, high intake of UPF was associated with high energy intake. Men with the highest intake of UPF were less likely to be current drinkers and smokers, but women were more likely to be current drinkers. Regular physical exercise was less prevalent in men and more prevalent in women with high consumption of UPF. Furthermore, the highest consumption of UPF was related to a high BMI in men but low BMI in women. On the other hand, participants with at least one chronic disease, and menopausal women were less likely to be high consumers of UPF (Table 1). The detailed categories of categorical variables are shown in S2 Table.

The most important sources of UPF were yoghurt, sweet rice punch, breads, and soymilk drinks in women, but yoghurt, soft drinks and fruit sodas, soymilk drink, sweet rice punch, instant noodles, and breads in men (S3 Fig).

Fig 2 shows the HRs and 95% CI for the risk of mortality according to quartiles of UPF consumption. After adjusting for potential confounders (model 4), the HRs (95% CI) of all-cause mortality for the highest (Q4) vs lowest (Q1) intake of UPF were 1.08 (0.85–1.12) in men and 0.95 (0.81–1.11) in women. Regarding cause-specific mortality, the HR (95% CI) for cancer and CVD mortality, respectively, were 1.02 (0.84–1.22) and 0.88 (0.64–1.22) in men, and 1.02 (0.83–1.26) and 0.80 (0.53–1.19) in women. The details of the derivation of final models shown in S3 Table.

We did not find evidence of an overall dose-response association between UPF intake and all-cause mortality (men, P-value = 0.064; women P = 0.526) (Fig 3). Similarly, there was no evidence of a dose-response association between UPF intake and cancer- or CVD-related mortality (S1 and S2 Figs).

The association between the intake of UPF items/sub-groups and all-cause mortality are shown in Table 2 and S4 Table. When participants with the highest intake of UPF items/sub-groups were compared to those with the lowest intake, the intake of ultra-processed red meat and fish was positively associated with all-cause mortality (HR (95% CI): men, 1.26 (1.11–1.43; women, 1.22 (1.05–1.43). In addition, the risk of all-cause mortality was high among men with the highest consumption of ultra-processed milk (HR 1.13, 95% CI 1.01–1.26) and ultra-processed soymilk drink (HR 1.12, 95% CI 1.0–1.25). There was no evidence of an association between the intake of UPF items/subgroups and cause-specific mortality (S5 and S6 Tables).

**Table 1. Characteristics of participants according to quartiles of ultra-processed food intake.**

| Characteristic | Quartiles of UPF intake, % food weight | | | | | | | |
| | Men | | | | Women | | | |
| | Q1 | Q2 | Q3 | Q4 | Q1 | Q2 | Q3 | Q4 |
| --- | --- | --- | --- | --- | --- | --- | --- | --- |
| Participants, n | 9711 | 9712 | 9712 | 9712 | 18,682 | 18,682 | 18,683 | 18,682 |
| Age, years | 56.2±0.1 | 53.8±0.1 | 52.8±0.1 | 52.1±0.1 | 54.9±0.1 | 52.1±0.1 | 51.5±0.1 | 51.2±0.1 |
| Age group, 40–49 years | 21.3 | 33.1 | 37.5 | 41.8 | 25.6 | 38.8 | 41.7 | 43.7 |
| Education level, ≥ College | 33.2 | 40.2 | 45.6 | 48.5 | 14.0 | 21.9 | 26.6 | 29.3 |
| Monthly income, ≥3000USD | 39.3 | 46.0 | 50.7 | 51.1 | 31.2 | 40.9 | 43.6 | 44.5 |
| Marital status, married/cohabiting | 95.2 | 94.8 | 94.6 | 92.6 | 86.2 | 88.4 | 88.3 | 86.2 |
| Residence, Seoul/ Metropolitan | 34.0 | 36.6 | 38.1 | 39.6 | 30.5 | 33.3 | 35.7 | 37.5 |
| Current smoker | 42.7 | 43.0 | 40.2 | 38.7 | 0.9 | 1.2 | 1.3 | 1.6 |
| Current drinker | 74.3 | 74.2 | 72.3 | 70.2 | 25.1 | 32.5 | 32.6 | 33.0 |
| AMI, kcal/day | 1434 | 1342 | 1408 | 1360 | 1599 | 1582 | 1633 | 1599 |
| Regular physical activity | 36.8 | 35.3 | 36.1 | 35.2 | 34.9 | 35.0 | 36.5 | 37.3 |
| BMI, kg/m$^2$ | 24.4±0.0 | 24.4±0.0 | 24.4±0.0 | 24.4±0.0 | 23.7±0.0 | 23.7±0.0 | 23.6±0.0 | 23.4±0.0 |
| BMI category, ≥25.0kg/m$^2$ | 38.4 | 40.5 | 40.3 | 40.0 | 31.4 | 29.1 | 26.5 | 25.1 |
| Energy intake, kcal/day | 1672 | 1787 | 1904 | 2004 | 1525 | 1632 | 1745 | 1814 |
| UPF intake, median | 1.9 | 4.5 | 7.8 | 13.7 | 1.9 | 4.6 | 8.1 | 14 .0 |
| NOVA 1, median | 96.4 | 93.0 | 89.3 | 82.2 | 97.1 | 93.7 | 89.8 | 82.7 |
| Disease score, ≥2 | 9.0 | 7.8 | 7.9 | 7.9 | 7.5 | 6.3 | 6.2 | 6.7 |
| Postmenopausal women | | | | | 73.6 | 61.0 | 58.5 | 57.1 |
| Oral contraceptives user | | | | | 82.5 | 83.2 | 84.1 | 83.4 |

Values are % or mean ±SE unless indicated otherwise. Details are shown in S2 Table

The mean BMI was adjusted for age and the means of dietary variables were adjusted for age and total energy intake.

AMI, Total activity Metabolic Index; ALT, Alanine aminotransferase; BMI, body mass index; MI, myocardial infarction; CPD, chronic pulmonary diseases; CKD, chronic kidney disease, NOVA1, Unprocessed/minimally processed foods.

Table 3 shows the association between UPF intake and all-cause mortality stratified by demographic, lifestyle and clinical characteristics in men and women. There was no evidence of interaction effects between participant characteristics and UPF intake on all-cause mortality (all FDR-adjusted P>0.05).

In the sensitivity analyses, the HR were consistent even after excluding participants who died within the first 5-years of follow-up, those who died from accidents, those with missing data on confounding variables, and after adjusting for LTPA and alcohol intake (g/d). However, when we adjusted for the proportion of unprocessed/minimally processed food intake, the HR was slightly reduced in men and increased in women (S7 Table).

## Discussion

The analysis of a large population-based cohort from an Asian population found no evidence of an association of total UPF consumption with all-cause or cause-specific mortality but reported a positive association of ultra-processed red meat and fish in both men and women, and ultra-processed milk and soymilk drinks in men, with all-cause mortality.

The lack of association between total UPF and all-cause mortality contrasts with previous findings. However, previous studies are mostly from European and North American countries whose overall dietary patterns are different from those of Asian countries [36]. Kim and colleagues used FFQ data from the National Health and Nutrition Examination survey (NHANES) to assess the association between UPF intake and 19-year risk of all-cause and

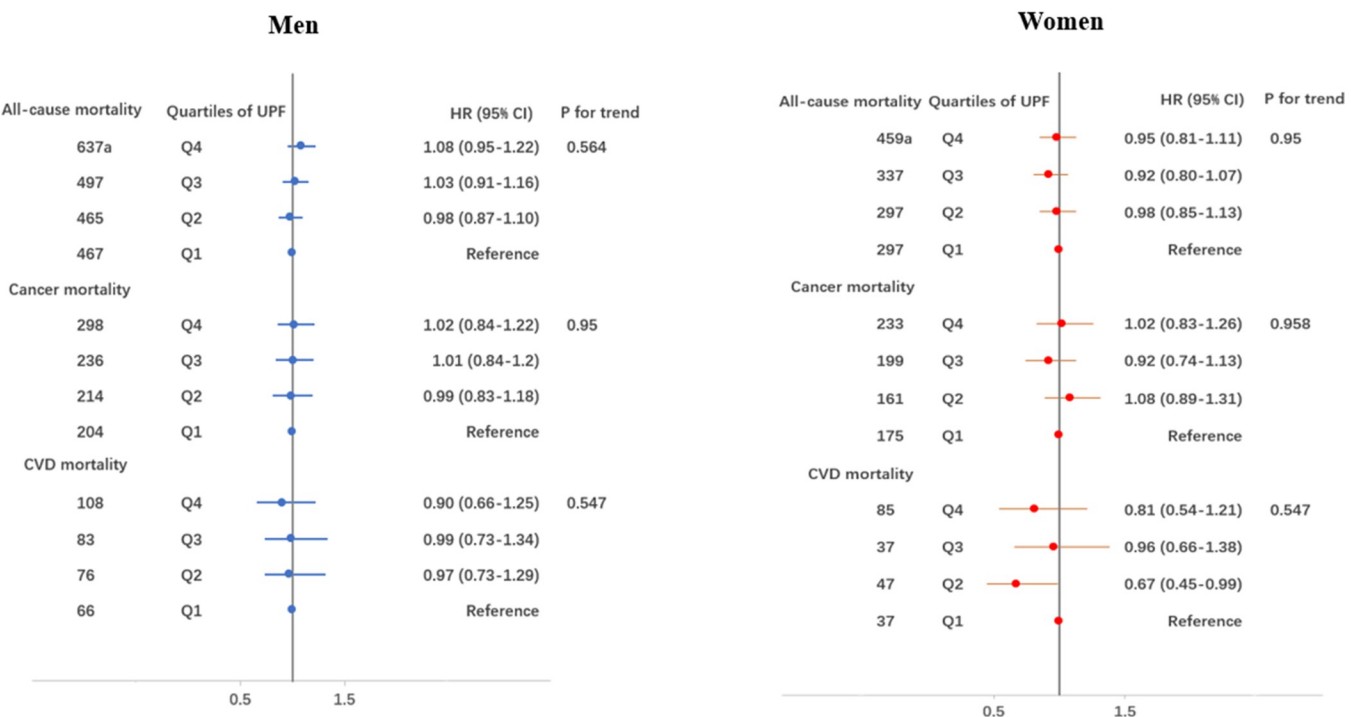

**Fig 2. Hazard ratios and 95% CI of all-cause, cancer and CVD mortality according to quartiles of UPF intake in men and women.** [a]Events. Hazard ratios were sequentially adjusted for age, total energy intake, education level, monthly income, marital status, smoking, drinking, regular physical exercise, BMI, disease score, menopausal status, use of oral contraceptives and hypercholesterolemia (for CVD mortality). Detailed models are shown in supplementary material.

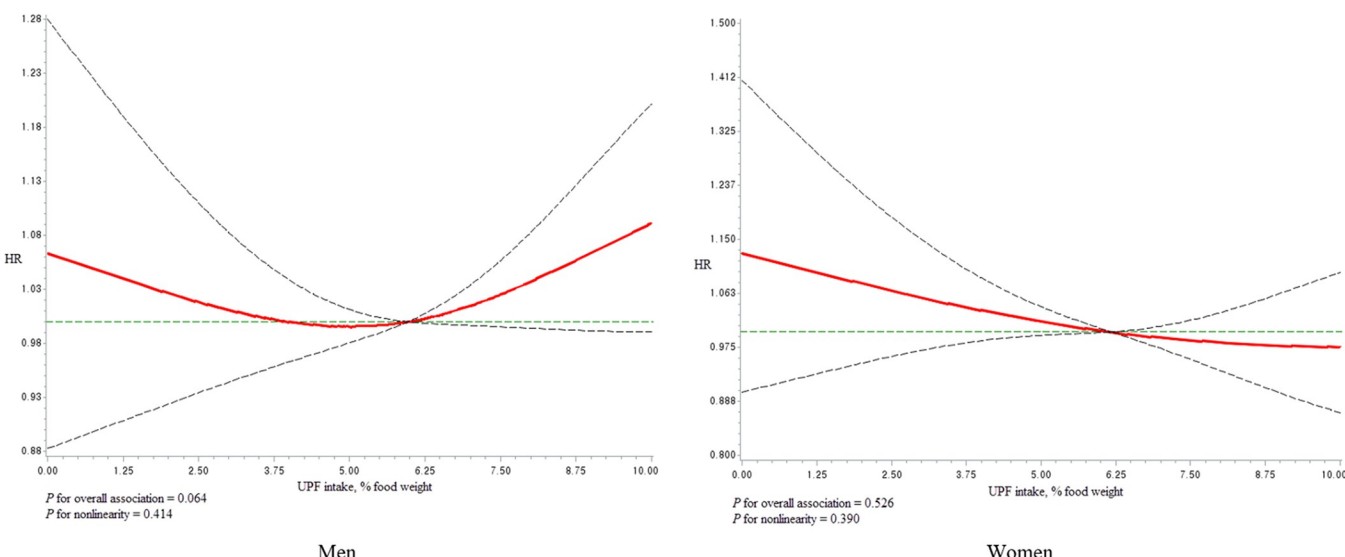

**Fig 3. Dose-response association of UPF intake and all-cause mortality in men and women using regression splines with three knots at the 5th, 50th and 95th percentiles of the UPF distribution.** The red line indicate hazard ratios (HR) and the dotted lines indicate the 95% CI. The reference value for HRs was 5.6 (median of % food weight from UPF). The model was adjusted for age, total energy intake, education level, monthly income, marital status, smoking, drinking, regular physical exercise, comorbidity score, menopausal status, and use of oral contraceptives.

**Table 2. Association of UPF items/ subgroups and all-cause mortality.**

| | | Men | Women |
|---|---|---|---|
| | Lowest quartile | Highest quartile | Highest quartile |
| Ultra-processed food items | HR (95% CI)[1] | HR (95% CI) | HR (95% CI) |
| Instant noodles | 1.00 | 1.04 (0.92–1.18) | 1.05 (0.92–1.21) |
| Breads | 1.00 | 0.96 (0.85–1.08) | 1.05 (0.9–1.23) |
| Bread spreads | 1.00 | 1.02 (0.92–1.13) | 0.94 (0.82–1.07) |
| Breakfast cereals & snacks | 1.00 | 0.96 (0.87–1.07) | 1.13 (0.47–2.75) |
| Candies and chocolate | 1.00 | 0.95 (0.86–1.04) | 0.92 (0.81–1.04) |
| Pizza and hamburger | 1.00 | 0.93 (0.82–1.06) | 0.93 (0.80–1.10) |
| Red meat and fish | 1.00 | 1.26 (1.11–1.43) | 1.22 (1.05–1.43) |
| Milk | 1.00 | 1.13 (1.01–1.26) | 1.08 (0.93–1.25) |
| Yoghurt | 1.00 | 1.05 (0.93–1.18) | 0.88 (0.77–1.02) |
| Ice cream | 1.00 | 0.94 (0.83–1.06) | 0.81 (0.69–0.94) |
| Coffee creamer | 1.00 | 1.00 (0.87–1.14) | 0.92 (0.81–1.04) |
| Soymilk drink | 1.00 | 1.12 (1.00–1.25) | 1.08 (0.96–1.21) |
| Soft drinks & fruit sodas | 1.00 | 0.92 (0.99–1.10) | 1.01 (0.88–1.16) |
| Sweet rice punch (*Sikhye*) | 1.00 | 0.95 (0.84–1.07) | 0.93 (0.81–1.07) |

1 Adjusted for age, total energy intake, education level, monthly income, marital status, smoking, drinking, regular physical exercise, BMI, comorbidity score, menopausal status, and use of oral contraceptives. Bread spreads included jam, margarine, and butter. The details are shown in supplementary material.

CVD-mortality. Frequent consumption of UPF increased the risk of all-cause mortality by 31%, but not CVD mortality [9]. However, this study assessed frequency of UPF intake rather than its absolute intake. Another study from Spain used a computerized dietary history questionnaire to compare the mortality risk of the highest versus lowest quartiles of UPF consumption (g/d and percent of energy) and reported that highest consumption of UPF was associated with increased risk of total mortality [10]. Another Spanish study of healthy university graduates (20–91 years) reported that consumption of >4 daily servings of UPF was associated with a 62% increased risk of all-cause mortality [11]. This study comprised young university graduates, which limited its external validity.

Zhong et al investigated the relationship between UPF intake (in serving sizes) and CVD mortality among healthy subjects who were recruited for cancer screening. The risk of death among subjects in the highest quintiles of daily servings of UPF were 1.50, and 1.68 for CVD and CHD-related mortality, respectively. This study comprised non-Hispanic whites and highly educated subjects, which limited its external validity [8], and used serving sizes which may not reflect the actual consumption of UPF. Romero et al followed 5-60-year-old healthy Spaniards for 27 years and reported a 15% increase in the risk of all-cause mortality among highest consumers of UPF [12]. A France-based study recruited 44,551 healthy participants aged ≥45 years of whom 73.3% were women. This study used a web-based 24-hour recall to estimate UPF intake, and UPF contributed 29% to total energy intake. After a median follow-up of 7.1 years. A 14% increased risk of all-cause mortality occurred for each 10% increase in energy consumption from UPF. However, participants in this study were volunteers and more health conscious, which might have underestimated the true association in the general population [13]. An Italian study also reported an increased risk of all-cause mortality among consumers of high UPF, but not cancer-specific mortality [14]. A systematic review and meta-analysis of cohort studies reported high consumption of UPF might increase the risk of all-cause mortality [37].

**Table 3. Association of total UPF intake and all-cause mortality stratified by participant characteristics.**

| | | Quartiles of UPF intake (% of food weight) | | | | | | | | P for trend | P for interaction* |
|---|---|---|---|---|---|---|---|---|---|---|---|
| | | Q1 | | Q2 | | Q3 | | Q4 | | | |
| Characteristic | Strata | Deaths | HR (95% CI)[1] | Deaths | HR (95% CI) | Deaths | HR (95% CI) | Deaths | HR (95% CI) | | |
| Age group | 40–49 | 91 | 1.00 | 116 | 0.89 (0.68–1.18) | 130 | 0.93 (0.72–1.22) | 142 | 0.94 (0.72–1.23) | 0.890 | |
| | 50–59 | 309 | 1.00 | 281 | 1.01 (0.88–1.23) | 267 | 1.04 (0.88–1.23) | 245 | 0.98 (0.83–1.17) | 0.890 | 0.930 |
| | 60–69 | 696 | 1.00 | 437 | 0.95 (0.85–1.10) | 365 | 0.92 (0.81–1.05) | 375 | 1.04 (0.91–1.18) | 0.800 | |
| Education | Elementary | 358 | 1.00 | 203 | 0.98 (0.82–1.16) | 142 | 0.89 (0.73–1.08) | 140 | 0.99 (0.81–1.21) | 0.990 | |
| | Middle school | 211 | 1.00 | 164 | 1.05 (0.86–1.30) | 139 | 1.09 (0.87–1.36) | 124 | 1.05 (0.84–1.32) | 0.660 | |
| | High school | 358 | 1.00 | 296 | 0.95 (0.82–1.11) | 307 | 1.03 (0.88–1.20) | 297 | 1.05 (0.9–1.23) | 0.370 | 0.930 |
| | College and above | 169 | 1.00 | 171 | 0.98 (0.79–1.22) | 174 | 0.93 (0.75–1.16) | 201 | 1.00 (0.81–1.23) | 0.980 | |
| Income, USD | <1000 | 286 | 1.00 | 140 | 0.79 (0.64–0.96) | 130 | 0.94 (0.76–1.17) | 123 | 0.94 (0.75–1.16) | 0.770 | |
| | 1000–3000 | 432 | 1.00 | 400 | 1.17 (1.02–1.35) | 306 | 1.01 (0.87–1.17) | 323 | 1.11 (0.95–1.29) | 0.504 | 0.522 |
| | ≥3000 | 248 | 1.00 | 215 | 0.87 (0.72–1.04) | 252 | 0.98 (0.82–1.18) | 237 | 0.99 (0.82–1.19) | 0.700 | |
| Marital status | Married/cohabiting | 145 | 1.00 | 94 | 1.00 (0.91–1.10) | 103 | 0.97 (0.88–1.07) | 123 | 1.01 (0.91–1.12) | 0.601 | 0.695 |
| | Single/others | 951 | 1.00 | 740 | 0.85 (0.66–1.11) | 659 | 1.05 (0.81–1.36) | 639 | 1.08 (0.84–1.39) | 0.456 | |
| Drinking | Never | 514 | 1.00 | 359 | 0.99 (0.87–1.14) | 314 | 0.92 (0.80–1.06) | 317 | 0.95 (0.82–1.10) | 0.400 | |
| | Current | 488 | 1.00 | 413 | 1.04 (0.91–1.19) | 384 | 1.09 (0.95–1.25) | 356 | 1.09 (0.94–1.26) | 0.220 | 0.522 |
| | Past | 94 | 1.00 | 62 | 0.69 (0.50–0.96) | 064 | 0.83 (0.60–1.15) | 89 | 1.08 (0.79–1.47) | 0.230 | |
| Smoking | Never | 589 | 1.00 | 411 | 0.93 (0.82–1.06) | 379 | 0.91 (0.80–1.04) | 393 | 0.99 (0.86–1.13) | 0.870 | |
| | Current | 257 | 1.00 | 215 | 1.02 (0.85–1.22) | 204 | 1.15 (0.95–1.39) | 181 | 1.06 (0.87–1.29) | 0.451 | 0.930 |
| | Past | 250 | 1.00 | 208 | 1.05 (0.87–1.26) | 179 | 0.99 (0.81–1.21) | 188 | 1.08 (0.88–1.32) | 0.560 | |
| Regular exercise | No | 747 | 1.00 | 570 | 0.98 (0.88–1.1) | 498 | 0.96 (0.86–1.08) | 500 | 1.03 (0.91–1.16) | 0.672 | |
| | Yes | 349 | 1.00 | 264 | 0.98 (0.83–1.15) | 264 | 1.02 (0.87–1.20) | 262 | 1.02 (0.87–1.21) | 0.701 | 0.930 |
| BMI category | Underweight | 28 | 1.00 | 32 | 1.18 (0.69–2.01) | 16 | 0.88 (0.46–1.67) | 27 | 1.30 (0.73–2.29) | 0.483 | |
| | Normal weight | 376 | 1.00 | 280 | 0.98 (0.84–1.15) | 286 | 1.07 (0.91–1.25) | 274 | 1.02 (0.87–1.20) | 0.656 | |
| | Overweight | 315 | 1.00 | 216 | 0.93 (0.78–1.11) | 204 | 0.95 (0.79–1.14) | 203 | 1.00 (0.83–1.20) | 0.932 | 0.930 |
| | Obese | 377 | 1.00 | 306 | 1.00 (0.86–1.17) | 256 | 0.94 (0.80–1.11) | 258 | 1.04 (0.88–1.23) | 0.733 | |
| Disease score | 0 | 818 | 1.00 | 649 | 1.01 (0.91–1.12) | 570 | 0.98 (0.88–1.09) | 588 | 1.06 (0.95–1.19) | 0.330 | 0.930 |

(*Continued*)

**Table 3.** (Continued)

| Characteristic | Strata | Quartiles of UPF intake (% of food weight) | | | | | | | | P for trend | P for interaction* |
|---|---|---|---|---|---|---|---|---|---|---|---|
| | | Q1 | | Q2 | | Q3 | | Q4 | | | |
| | | Deaths | HR (95% CI)[1] | Deaths | HR (95% CI) | Deaths | HR (95% CI) | Deaths | HR (95% CI) | | |
| | ≥1 | 278 | 1.00 | 185 | 0.90 (0.74–1.08) | 192 | 1.01 (0.83–1.22) | 174 | 0.93 (0.77–1.13) | 0.699 | |

1 Adjusted for age, total energy intake, education level, monthly income, marital status, smoking, drinking, regular physical exercise, BMI, comorbidity score, menopausal status, and use of oral contraceptives.

*FDR adjusted

Previous studies were conducted in the US [8, 9], Spain [10–12], France [13] and Italy [14]. The dietary assessment methods, heterogeneity of populations studied, and the different composition of UPF foods in these countries could explain the observed differences. The contribution of UPF to the diet of the Korean population was 25.1% in terms of total energy [18], which is low compared to 45% in Canada [38], 56.8% in the UK [39], 57.9% in the US [40] and 29% France [13]. This suggests that Korean dietary patterns, although at a slow pace, are tending towards Western dietary patterns. Projections indicate that the sales of UPF in Southeast and East Asian countries will approach those of high-income countries by 2035 [1]. The median percentage weight of UPF in the current study was 5.6%, which is lower than that reported in a previous study in Korea [18]. The previous study estimated UPF intake from 24-hour recall data, expressed UPF intake as a percentage of total energy, included distilled alcoholic beverages in the calculation of UPF, and used dietary data from 2016–2018 which reflects current intake. Nevertheless, the low intake of UPF in this population reflects unique dietary patterns in Korea and could be one of the reasons for the observed association between UPF intake and mortality. Although Korea has undergone rapid socioeconomic transformation, the traditional dietary pattern characterized by high intake of vegetables and low intake of fat has been retained in this population. This is due to efforts by Government and private organizations to promote healthy eating, and the conservative nature of Korean people towards traditional foods and cooking styles [41]. The unique aspects of the Korean diet are also notable. For example, sodium additives constitute two thirds of total daily salt intake in the Western diet, present in UPF [40]. However, the major sources of sodium in the Korean diet are *kimchi* (fermented vegetables), fermented soybean paste and cooked dishes [42, 43]. Furthermore, meat in Korea is dominantly Korean-style cooked meat, consumed as soups, grilled marinated beef, and grilled pork belly, typically wrapped in various vegetables. The consumption of Western style-cooked meat (processed red meat) is low in this population [41].

It is possible that total UPF intake is not associated with mortality in the Korean population. The nutrition, demographic, and epidemiological transition in Asian countries has been accompanied by changes in lifestyle such as smoking, drinking, physical activity, and improvement in the health system, which strongly influence mortality than dietary factors [44]. High consumption of UPF was associated with high income, high education, and urban residence suggesting that UPF consumption is associated with high socio-economic status in this population, which may be linked to protective factors against mortality [44, 45]. Even though we stratified the analyses by demographic and clinical variables, we cannot rule out residual confounding and additional confounders for which we could not adjust.

Nevertheless, we found a 26% increase in the risk of all-cause mortality among highest consumers of ultra-processed red meat and fish. Processed red meat has been associated with

mortality in Western countries [37, 46]. In a pooled analysis of data from Asian populations, total meat intake was not associated with mortality, but the intake of red meat and poultry was inversely associated with CVD and cancer mortality [45]. However, this study did not differentiate between unprocessed and processed red meat, which might explain the difference observed in this study. Iron mutagens generated by high-temperature cooking [47, 48], and N-nitroso compounds formed in processed meat and endogenously from heme iron [49, 50] are some of the mechanisms that explain the detrimental health impacts of processed red meat on health. In animal studies, metabolism by intestinal microbiota of dietary L-carnitine, a trimethylamine abundant in red meat, also produces trimethylamine oxide (TMAO) and accelerates atherosclerosis [51]. Ultra-processed red meats are also high in sodium additives in form of sodium chloride, monosodium glutamate (flavor enhancer), sodium phosphates (water-binding), sodium nitrite/nitrate (preservative and color stabilizer), sodium lactate (preservative and antioxidant) and sodium erythorbate (reducing agent). Sodium intake is associated with CVD risk and various health outcomes [7].

Ultra-processed, flavored, sugar-sweetened milk, and soymilk drink intake was associated with an elevated risk of mortality in men, consistent with a recent meta-analysis that found positive associations of sugar-sweetened beverages, artificially sweetened beverages, with all-cause mortality [37]. These products are high in added sugars and dietary cholesterol. Although we did not investigate potential compounds that explained the milk-mortality association, Bonaccio et al. reported that 23% of the association between UPF intake and total mortality was explained by sugar and 9% by dietary cholesterol which are all milk components. Previous studies have also suggested that the highest proportion of detrimental health effects of UPF may be explained by factors other than nutrient composition [9, 11, 14].

Longitudinal studies that assess the mediating role of non-nutrient UPF compounds on mortality are necessary to confirm whether the associations between UPF and health outcomes are indeed due to processing methods. The components added to foods during ultra-processing are believed to promote chronic diseases via diet-microbiome-host interactions [52–54]. These components include cosmetic food additives such as coloring agents, emulsifiers, and preservatives, which are used to enhance the flavor and palatability of UPF products [24, 55]. Food processing also results in the formation of substances such as trans-fatty acids, furans, and contaminants [55], and the consumption of UPF may be associated with increased exposure to endocrine-disrupting chemicals [56], which have been identified as contributors to adverse health outcomes [57]. Ultra-processing also results in the alteration and recombination of the food matrix, and results in food products with a high glycemic index than minimally processed foods [58].

Besides the strengths of this study such as a large sample size from a population-based survey which increases generalizability to the Korean population, inclusion of a yet under-studied population, the prospective design, analysis of specific UPF items, stratification by potential demographic, lifestyle and clinical confounders, use of a validated SQFFQ, and conducting several sensitivity analyses, several limitations should be acknowledged. First, this was an observational study based on adults prone to chronic diseases. This limits the confirmation of causal relationships and generalizability of our findings to young individuals. Second, although we adjusted for potential covariates and conducted stratified analyses, there is still a possibility of unmeasured and residual confounding. A single measure of dietary intake was used; however, UPF intake could have changed over time, resulting in underestimation of the true association between UPF intake and mortality because of non-differential misclassification. Third, the FFQ used in this study was not designed to assess UPF intake, and the identification of specific UPF items from dishes was limited, thus underestimation of UPF intake is possible. The NOVA classification has been criticized as ambiguous with an inherent risk of

misclassification owing to its purely descriptive nature [59]. For example, Braesco and colleagues reported low consistency of NOVA food classification among different experts even in the presence of food ingredient information. The ambiguity could arise from the indication that UPF are "industrial formulations with many ingredients" or processing procedures. Unprocessed/minimally processed foods such as commercial juices could be perceived as industrial in nature and therefore classified as UPF by some evaluators [59]. Furthermore, the definition of UPF has changed over time, which could reduce the consistency of UPF definition and examples of UPF foods [60]. In this study, un-processed foods such as yoghurt, and processed foods such as bakery-made breads could have been classified as ultra-processed foods since we lacked detailed processing information from the SQFFQ. Potential misclassification could have resulted in an inaccurate estimation of the risk of mortality towards the null. Nevertheless, we employed a new approach that could be used in large cohort studies to minimize the misclassification of UPF [25]. Using this approach, Khandpur and colleagues 'doubted' only 4.4% of the 205 food items. Finally, dietary data were self-reported, which increases the possibility of underreporting of the true intake of UPF.

## Conclusion

Data from a large population-based cohort study in Korea found no evidence that a higher intake of total ultra-processed foods is associated with mortality. Ultra-processed meat and fish were positively associated with all-cause mortality in both men and women, whereas ultra-processed milk and soymilk drink intake was positively associated with all-cause mortality in men. Further longitudinal studies from diverse populations are needed to strengthen the evidence of potential detrimental effects of UPF on health. In the future, the unique dietary and demographic features of the Korean population that underlie the observed associations in this study should be investigated.

## Supporting information

**S1 Table. Classification of FFQ food items according to the NOVA classification.**
(DOCX)

**S2 Table. Characteristics of participants according to quartiles of ultra-processed food intake.**
(DOCX)

**S3 Table. Hazard ratios and 95% CI of all-cause, cancer and CVD mortality according to quartiles of UPF intake.**
(DOCX)

**S4 Table. Association of UPF items/ subgroups and all-cause mortality.**
(DOCX)

**S5 Table. Association of ultra-processed food items/ subgroups and cancer-specific mortality.**
(DOCX)

**S6 Table. Association of ultra-processed food items/subgroups and CVD-specific mortality.**
(DOCX)

**S7 Table. Sensitivity analyses of the association between total UPF intake and all-cause mortality.**
(DOCX)

**S1 Fig. Dose-response association of UPF intake and cancer mortality using regression splines with three knots at the 5th, 50th and 95th percentiles of UPF distribution.** The red line indicates hazard ratios (HR), and the dotted lines indicate the 95% CI. The reference value for HRs was 5.6 (median of % food weight from UPF). The model was adjusted for age, total energy intake, education level, monthly income, marital status, smoking, drinking, regular physical exercise, comorbidity score, menopausal status, and use of oral contraceptives. (TIF)

**S2 Fig. Dose-response association of UPF intake and CVD mortality using regression splines with three knots at the 5th, 50th and 95th percentiles of UPF distribution.** The red line indicates hazard ratios (HR), and the dotted lines indicate the 95% CI. The reference value for HRs was 5.6 (median of % food weight from UPF). The model was adjusted for age, total energy intake, education level, monthly income, marital status, smoking, drinking, regular physical exercise, comorbidity score, menopausal status, use of oral contraceptives, and hyper-cholesterolemia. (TIF)

**S3 Fig. Median contribution of UPF items/sub-groups to total UPF intake.** (TIF)

## Acknowledgments

We thank the participants in the HEXA cohort for their valuable contribution to this cohort.

## Author Contributions

**Conceptualization:** Anthony Kityo.

**Formal analysis:** Anthony Kityo.

**Investigation:** Anthony Kityo.

**Methodology:** Anthony Kityo, Sang-Ah Lee.

**Project administration:** Sang-Ah Lee.

**Resources:** Sang-Ah Lee.

**Supervision:** Sang-Ah Lee.

**Validation:** Anthony Kityo, Sang-Ah Lee.

**Visualization:** Anthony Kityo.

**Writing – original draft:** Anthony Kityo.

**Writing – review & editing:** Sang-Ah Lee.

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
