## [Decision Letter · Decision Letter 0]

16 Mar 2023

PONE-D-22-26654The Intake of Ultra-processed foods, All-cause, Cancer and Cardiovascular Mortality in the Health Examinees (HEXA) CohortPLOS ONE

Dear Dr. Lee,

Thank you for submitting your manuscript to PLOS ONE. After careful consideration, we feel that it has merit but does not fully meet PLOS ONE’s publication criteria as it currently stands. Therefore, we invite you to submit a revised version of the manuscript that addresses the points raised during the review process.

It is a significant study where authors investigated the association between the consumption of ultra-processed foods and cardiovascular and total mortality in a large cohort study. However, some comments are raised by the reviewers. Authors are suggested to address them carefully, and improved the manuscript further.

We look forward to receiving your revised manuscript.

Kind regards,

Shaonong Dang, PhD

Academic Editor

PLOS ONE

3. Thank you for stating the following in the Acknowledgments/ funding Section of your manuscript:

“The Korean Genome Epidemiology project was initiated by the Korean national Research Institute of Health (NIH), Centers for Disease Control and Prevention and the Ministry of Health and Welfare with funding from the Korean Government [grant number 2004-E71004-00; 2005-E71011-00; 2005-E71009-00; 2006-E71001-00; 2006-E71004-00; 2006-E71010-00; 2006E71003-00; 2007-E71004-00; 2007-E71006-00; 2008-E7100600; 2008-E71008-00; 2009-E71009-00; 2010-E71006-00; 2011E71006-00; 2012-E71001-00; 2013-E71009-00]. The funding sources had no role in the design, execution, analyses, and interpretation of the data or decision to submit the results of this study.”

“The Korean Genome Epidemiology project was initiated by the Korean national Research Institute of Health (NIH), Centers for Disease Control and Prevention and the Ministry of Health and Welfare with funding from the Korean Government [grant number 2004-E71004-00; 2005-E71011-00; 2005-E71009-00; 2006-E71001-00; 2006-E71004-00; 2006-E71010-00; 2006E71003-00; 2007-E71004-00; 2007-E71006-00; 2008-E7100600; 2008-E71008-00; 2009-E71009-00; 2010-E71006-00; 2011E71006-00; 2012-E71001-00; 2013-E71009-00]. The funding sources had no role in the design, execution, analyses, and interpretation of the data or decision to submit the results of this study.”

Additional Editor Comments:

It is a significant study where authors investigated the association between the consumption of ultra-processed foods and cardiovascular and total mortality in a large cohort study. However, some comments are raised by the reviewers. Authors are suggested to address them carefully, and improved the manuscript further.

Reviewers' comments:

Reviewer's Responses to Questions

**Comments to the Author**

1. Is the manuscript technically sound, and do the data support the conclusions?

Reviewer #1: Yes

Reviewer #2: Yes

2. Has the statistical analysis been performed appropriately and rigorously? 

Reviewer #1: Yes

Reviewer #2: Yes

3. Have the authors made all data underlying the findings in their manuscript fully available?

Reviewer #1: Yes

Reviewer #2: Yes

4. Is the manuscript presented in an intelligible fashion and written in standard English?

Reviewer #1: Yes

Reviewer #2: Yes

5. Review Comments to the Author

Reviewer #1: In the article authors investigate the association between the consumption of ultra-processed foods and cardiovascular and total mortality in a large cohort study. The article is very well-written. Although similar studies were conducted in the past, they were done mostly in the western countries and there is a paucity of data from Asia on the topic. Therefore I think it is worth to conduct such analysis, besides as authors emphasize, dietary habits in Korea and other Asian countries are changing towards more processed and ‘westernized’. It may be interesting to analyze these changes along with the impact on the health status.

Methodology was well explained. The biggest limitation of the analysis concerns usage of the single FFQ to assess the UPFs, additionally the FFQ was not developed to especially target UPFs. Nevertheless, the authors discuss these limitations. Inclusion of several adjusted models also decreases the risk of confounding.

In my opinion, the authors should discuss more thoroughly the difficulties and ambiguity of UPF definition. The NOVA classification has been also criticized in the past as the inaccurate tool. I think it can be mentioned in the discussion how it could have contributed to the results. Authors present that one of the biggest contributors to UPF was e.g. yogurt and bread. Not all breads and yogurts are ultra-processed. I agree that packaged bread and bread rolls with a lot of food additives to increase the shelf life are undoubtedly UPFs, on the other hand there are breads with much shorter list of ingredients with short shelf life available at bakeries. Similarly there are yogurt-like products which are flavored and heavily processed and also yogurts which contain three ingredients. I think it should be explained a bit better which characteristics of these products make them UPFs and not just PFs (perhaps as a footnote to the tables in supplementary material). Food processing differs between countries, therefore I think that inclusion of such details could be useful for the international reader. Misclassification of products could have contributed to the results.

Taken together, despite perhaps these minor changes, I recommend the article for publication in the journal.

Reviewer #2: The current manuscript technically sound. The data supported the conclusion. It is a very interesting study. And provid the evidence on this topic. Author had well orgnized and analyzed. There are some minor english grammer and spell have to check.

6. PLOS authors have the option to publish the peer review history of their article (what does this mean?). If published, this will include your full peer review and any attached files.

Reviewer #1: No

Reviewer #2: No

---

## [Author Response · Author response to Decision Letter 0]

13 Apr 2023

23 March 2023

Shaonong Dang, PhD

Academic Editor

PLOS ONE

Dear Dr. Dang, 

We have kindly enclosed the responses to editorial and reviewers’ comments, and the revised version of our manuscript ‘PONE-D-22-26654: The Intake of Ultra-processed foods, All-cause, Cancer and Cardiovascular Mortality in the Health Examinees (HEXA) Cohort’.

We sincerely thank the Editorial team, and the reviewers for considering our manuscript for publication, and for the valuable comments. All comments from the Editor and the Reviewers have been carefully addressed and changes have been highlighted in red throughout the marked manuscript text. The Editorial and Reviewers’ comments have been responded to.

We have discussed the ambiguities surrounding the NOVA classification system, and how potential food misclassifications arising from these ambiguities could have influenced our results.

The formatting of the manuscript was addressed to match with journal requirements, and minor grammatical errors were corrected. To improve the presentation of our results, we have reduced results in Table 1 and 2 (Table 3 in the original version) and presented detailed results in supplementary material. Table 2 of the original manuscript was converted to Figure 2 in the revised manuscript, and details presented in supplementary Tables (Table S2). Furthermore, we slightly modified the title of our study to comply with the new guideline of the Korean Genome and Epidemiology Study.

We hope that our modifications adequately address the Editor and Reviewers’ comments and that our paper now meets the publication criteria of PLOS ONE.

We thank you again for your consideration.

Sang-Ah Lee, PhD

Department of Preventive Medicine, Kangwon National University

sangahlee@kangwon.ac.kr

1. EDITORIAL COMMENTS

Date: Mar 16 2023 07:46PM

To: "Sang-Ah Lee" sangahlee@kangwon.ac.kr

From: "PLOS ONE" plosone@plos.org

Subject: PLOS ONE Decision: Revision required [PONE-D-22-26654]

PONE-D-22-26654

The Intake of Ultra-processed foods, All-cause, Cancer and Cardiovascular Mortality in the Health Examinees (HEXA) Cohort

PLOS ONE

Dear Dr. Lee,

Thank you for submitting your manuscript to PLOS ONE. After careful consideration, we feel that it has merit but does not fully meet PLOS ONE’s publication criteria as it currently stands. Therefore, we invite you to submit a revised version of the manuscript that addresses the points raised during the review process.

It is a significant study where authors investigated the association between the consumption of ultra-processed foods and cardiovascular and total mortality in a large cohort study. However, some comments are raised by the reviewers. Authors are suggested to address them carefully and improved the manuscript further.

We look forward to receiving your revised manuscript.

Kind regards,

Shaonong Dang, PhD

Academic Editor

PLOS ONE

Response: We formatted the manuscript headings (bold, with font size 18, 16, and 14points for level 1-level 3 subheadings respectively). We indented the first line of each paragraph throughout the manuscript. The doi’s were also included in our references in the ‘Manuscript’ file. We named Tables, Figures and supporting information according to style guidelines. We cited Figures as ‘Fig’ followed by figure number. We have submitted supporting Tables and Figures as separate files (S1-S7 Tables; and S1-S3 Figures). We removed figures from the main manuscript and submitted them as PACE-generated ‘tif’ files. We deleted funding information, conflict of interest information and author contribution from the main manuscript.

Response: We provided funding information for the Korean Genome Project. However, the conduct of our study was not funded. Accordingly, we would like to state that “The authors have no support or funding to report” in the funding information section.

3. Thank you for stating the following in the Acknowledgments/ funding Section of your manuscript:

“The Korean Genome Epidemiology project was initiated by the Korean national Research Institute of Health (NIH), Centers for Disease Control and Prevention and the Ministry of Health and Welfare with funding from the Korean Government [grant number 2004-E71004-00; 2005-E71011-00; 2005-E71009-00; 2006-E71001-00; 2006-E71004-00; 2006-E71010-00; 2006E71003-00; 2007-E71004-00; 2007-E71006-00; 2008-E7100600; 2008-E71008-00; 2009-E71009-00; 2010-E71006-00; 2011E71006-00; 2012-E71001-00; 2013-E71009-00]. The funding sources had no role in the design, execution, analyses, and interpretation of the data or decision to submit the results of this study.”

“The Korean Genome Epidemiology project was initiated by the Korean national Research Institute of Health (NIH), Centers for Disease Control and Prevention and the Ministry of Health and Welfare with funding from the Korean Government [grant number 2004-E71004-00; 2005-E71011-00; 2005-E71009-00; 2006-E71001-00; 2006-E71004-00; 2006-E71010-00; 2006E71003-00; 2007-E71004-00; 2007-E71006-00; 2008-E7100600; 2008-E71008-00; 2009-E71009-00; 2010-E71006-00; 2011E71006-00; 2012-E71001-00; 2013-E71009-00]. The funding sources had no role in the design, execution, analyses, and interpretation of the data or decision to submit the results of this study.”

The funding information was deleted from the main manuscript.

Response: We would like to update the funding information as follows: “The authors have no support or funding to report”.

Response: We would like to update the data availability statement as follows:

Data from the Health Examinees (HEXA) study is part of the Korean Genome and Epidemiology Study (KoGES), conducted by Korea Disease Control and Prevention Agency (KDCA). The Health Examinees Study dataset used in our study was merged with the Central Cancer Registry (KCCR) data provided by National Cancer Center of Korea in a collaborative agreement. The dataset analyzed in this study is maintained and managed by the Division of Population Health Research at the National Institute of Health, Korean Disease Control and Prevention Agency. It contains personal data that may potentially be sensitive to the patients, even though researchers are provided with an anonymized dataset that excludes resident registration numbers. Accordingly, the minimal data set used in the current study could not be publicly shared by the authors due to legal restriction on sharing sensitive patient information. Researchers are required to submit ethics approval, and a detailed research plan to the KDCA. Upon approval, the researchers are required to physically visit the KCDA and conduct the analysis from the KoGES data analysis room at the KCDA in Osong, Chungcheong Province, Republic of Korea. However, if the analysis does not involve linkage to the cancer registry, virtual access to the anonymized data set can be granted. Other researchers may request access to the anonymized data by contacting the following individuals at the Division of Population Health Research, National Institute of Health, Korea Disease Control and Prevention Agency: Senior Staff Scientist Dr. Jung Hyun Lee (jaylee1485@korea.kr); Director Dr. Kyoungho Lee (khlee3789@korea.kr).

Response: Data from the Health Examinees (HEXA) study is part of the Korean Genome and Epidemiology Study (KoGES), conducted by Korea Disease Control and Prevention Agency (KDCA). The Health Examinees Study dataset used in our study was merged with the Central Cancer Registry (KCCR) data provided by National Cancer Center of Korea in a collaborative agreement. The dataset analyzed in this study is maintained and managed by the Division of Population Health Research at the National Institute of Health, Korean Disease Control and Prevention Agency. It contains personal data that may potentially be sensitive to the patients, even though researchers are provided with an anonymized dataset that excludes resident registration numbers. Accordingly, the minimal data set used in the current study could not be publicly shared by the authors due to legal restriction on sharing sensitive patient information. Researchers are required to submit ethics approval, and a detailed research plan to the KDCA. Upon approval, the researchers are required to physically visit the KCDA and conduct the analysis from the KoGES data analysis room at the KCDA in Osong, Chungcheong Province, Republic of Korea. However, if the analysis does not involve linkage to the cancer registry, virtual access to the anonymized data set can be granted. Other researchers may request access to the anonymized data by contacting the following individuals at the Division of Population Health Research, National Institute of Health, Korea Disease Control and Prevention Agency: Senior Staff Scientist Dr. Jung Hyun Lee (jaylee1485@korea.kr); Director Dr. Kyoungho Lee (khlee3789@korea.kr).

Response: We appreciate your kindness. Please find the updated data availability statement above.

Additional Editor Comments:

It is a significant study where authors investigated the association between the consumption of ultra-processed foods and cardiovascular and total mortality in a large cohort study. However, some comments are raised by the reviewers. Authors are suggested to address them carefully and improved the manuscript further.

We sincerely thank the Editor for considering our manuscript for publication. All comments from the Reviewers have been carefully addressed and changes have been highlighted in red throughout the marked manuscript text. A point-by-point response to the reviewers’ comments is provided below and is highlighted in blue color.

2. REVIEWER'S COMMENTS 

Reviewer's Responses to Questions

Comments to the Author

Reviewer #1: In the article authors investigate the association between the consumption of ultra-processed foods and cardiovascular and total mortality in a large cohort study. The article is very well-written. Although similar studies were conducted in the past, they were done mostly in the western countries and there is a paucity of data from Asia on the topic. Therefore I think it is worth to conduct such analysis, besides as authors emphasize, dietary habits in Korea and other Asian countries are changing towards more processed and ‘westernized’. It may be interesting to analyze these changes along with the impact on the health status.

1a. Methodology was well explained. The biggest limitation of the analysis concerns usage of the single FFQ to assess the UPFs, additionally the FFQ was not developed to especially target UPFs. Nevertheless, the authors discuss these limitations. Inclusion of several adjusted models also decreases the risk of confounding.

Response: Thank you for acknowledging the timely significance and contribution of our study to the current literature on this subject. We believe that more studies conducted in diverse populations will be crucial in developing a solid evidence base that could guide policy formulation around ultra-processed foods. In trying to generate new evidence, we acknowledge the limitations of our study, and the care that should be taken to interpret our findings. 

b. In my opinion, the authors should discuss more thoroughly the difficulties and ambiguity of UPF definition. The NOVA classification has been also criticized in the past as the inaccurate tool. I think it can be mentioned in the discussion how it could have contributed to the results. 

Response: The difficulties in defining ultra-processed foods using the NOVA classification system have been indeed pointed out in previous works. We discussed these difficulties the discussion section at lines 402-415, and the new approach that was suggested to improve UPF identification in cohort studies was highlighted at line 414-417 in the revised manuscript. We also discussed the potential misclassification bias that could have been introduced by the UPF classification at line 410-414.

c. Authors present that one of the biggest contributors to UPF was e.g. yogurt and bread. Not all breads and yogurts are ultra-processed. I agree that packaged bread and bread rolls with a lot of food additives to increase the shelf life are undoubtedly UPFs, on the other hand there are breads with much shorter list of ingredients with short shelf life available at bakeries. Similarly there are yogurt-like products which are flavored and heavily processed and also yogurts which contain three ingredients. I think it should be explained a bit better which characteristics of these products make them UPFs and not just PFs (perhaps as a footnote to the tables in supplementary material). Food processing differs between countries, therefore I think that inclusion of such details could be useful for the international reader. Misclassification of products could have contributed to the results.

Response: We added a footnote to S1 Table to describe the characteristics that qualified breads and yoghurts to be ultra-processed.

Taken together, despite perhaps these minor changes, I recommend the article for publication in the journal.

Response: We thank you so much for recommending our manuscript for publication in PLOS ONE.

Reviewer #2: The current manuscript technically sound. The data supported the conclusion. It is a very interesting study. And provid the evidence on this topic. Author had well orgnized and analyzed. There are some minor english grammer and spell have to check.

Response: Thank you so much for the positive energy. We addressed the grammatical and spelling errors.

---

## [Editor Report · Decision Letter 1]

20 Apr 2023

The Intake of Ultra-processed foods, All-cause, Cancer and Cardiovascular Mortality in the Korean Genome and Epidemiology Study-Health Examinees (KoGES-HEXA) Cohort

PONE-D-22-26654R1

Dear Dr. Lee,

We’re pleased to inform you that your manuscript has been judged scientifically suitable for publication and will be formally accepted for publication once it meets all outstanding technical requirements.

Kind regards,

Shaonong Dang, PhD

Academic Editor

PLOS ONE
---

## [Editor Report · Acceptance letter]

25 Apr 2023

PONE-D-22-26654R1 

The intake of ultra-processed foods, all-cause, cancer and cardiovascular mortality in the Korean Genome and Epidemiology Study-Health Examinees (KoGES-HEXA) cohort 

Dear Dr. Lee:

I'm pleased to inform you that your manuscript has been deemed suitable for publication in PLOS ONE. Congratulations! Your manuscript is now with our production department. 

Kind regards, 

on behalf of

Dr. Shaonong Dang 

Academic Editor

PLOS ONE